# Current Technologies in Depolymerization Process and the Road Ahead

**DOI:** 10.3390/polym13030449

**Published:** 2021-01-30

**Authors:** Yu Miao, Annette von Jouanne, Alexandre Yokochi

**Affiliations:** 1School of Resources and Environmental Engineering, East China University of Science and Technology, Shanghai 200237, China; 2School of Engineering and Computer Science, Baylor University, Waco, TX 76798, USA; Annette_VonJouanne@baylor.edu (A.v.J.); Alex_Yokochi@baylor.edu (A.Y.)

**Keywords:** plastic pollution, plastic depolymerization and recycling, plastic depolymerization challenges

## Abstract

Although plastic is considered an indispensable commodity, plastic pollution is a major concern around the world due to its rapid accumulation rate, complexity, and lack of management. Some political policies, such as the Chinese import ban on plastic waste, force us to think about a long-term solution to eliminate plastic wastes. Converting waste plastics into liquid and gaseous fuels is considered a promising technique to eliminate the harm to the environment and decrease the dependence on fossil fuels, and recycling waste plastic by converting it into monomers is another effective solution to the plastic pollution problem. This paper presents the critical situation of plastic pollution, various methods of plastic depolymerization based on different kinds of polymers defined in the Society of the Plastics Industry (SPI) Resin Identification Coding System, and the opportunities and challenges in the future.

## 1. Introduction

A world without plastics, or synthetic organic polymers, seems unimaginable today since it has become a major commodity on a global scale and has infiltrated almost every aspect of human life. Due to their low cost, ease of manufacture, versatility, and imperviousness to water, plastics are used in a multitude of products of different scales, and packaging is the most significant sector of use [1]. As a material for packaging, plastic allows companies to market effectively, design attractive-looking and appealing-feeling packages, and prevent loss from storage and transportation throughout the world. However, plastic packaging is often used only once before being thrown away, like grocery bags, drink bottles, straws, food wrappers, and toy packages, and it makes up more than 40% of all plastic trash [2].

### 1.1. Plastic Pollution

#### 1.1.1. Rapid Accumulation of Plastic Wastes

Plastic pollution has become one of the most pressing environmental issues, as the world’s ability to deal with plastic wastes is challenged by their rapid accumulation caused by their slow degradation rate, increasing sorting fee due to the complexity of the plastic products, and most importantly, little driving force from society. The rapid growth of the use and disposal of plastic materials has proved to be a burden to our ecosystem: from 2 million metric tons (MT) produced in 1950 to 359 MT generated in 2018, with a cumulative total of 8.3 billion MT as of 2018 [3]. Management of this enormous increase and quantity of plastic waste has been challenging, especially in the areas of rapid economic development and population growth. Between 1950 and 2015, approximately 8% of plastics ever produced have been incinerated, while only 7% have been recycled. Most of the produced plastics (around 55%) were discarded and are accumulating in landfills or the natural environment, while the rest are currently in use (see Figure 1) [3].

#### 1.1.2. Complexity of Plastic Wastes

The challenges posed by plastics recycling are much greater than for other products such as metal, glass, or paper. Plastics production is made up of about 70% thermoplastics and polyurethanes, 16% PP&A fibers and 14% thermosets, adhesives, coatings, and sealants. The latter two subsets are not recyclable using current technologies, and the adhesives and sealants are simply not collectable [4,5]. There are material properties that can limit the number of times that products can be recycled.

Bioplastics are usually touted as being eco-friendly. The often-cited advantages of bioplastics are reduced use of fossil fuel resources, a smaller carbon footprint, faster decomposition, and low toxicity. Not all bioplastics are biodegradable; in other words, they can be broken down completely into the water, carbon dioxide, and compost by microorganisms under the right conditions [6]. The biodegradable plastics only accounted for around 55% of the bioplastics production in 2019 (see Figure 2) [7]. Even the biodegradable plastics are blended to achieve commercially functional properties, so the environmental fate of these blends is unknown. Therefore, the blends need careful postconsumer management and further design to allow more rapid biodegradation before they can be safely released into the environment [8].

#### 1.1.3. Actions from Society 

Another reason causing the accumulation of plastic wastes is the lack of driving force to control and manage them. Plastic waste trade is a major means of dealing with insufficient recycling capacities. Since 1992, upward of half of the plastic waste intended for recycling has been exported to hundreds of countries around the world, and China is the leading importer who has imported a cumulative 45% of plastic waste [9]. The emerging markets in China found that the imported plastic waste could be used profitably and to manufacture more goods for sale or export. For exporting countries, shipping processed plastic waste to other countries has provided an outlet for plastic waste management, preventing it from accumulating in landfills or incineration in the source countries [10]. However, in 2017, China implemented a new policy banning the importation of most plastic wastes, forcing us to think about where to find a home for them (see Figure 3) [9,11]. 

A battle over plastic is also underway across the United States. Jurisdictions have instituted bans and extra charges on various types of plastics in California, New York, and hundreds of municipalities in the U.S. However, seventeen states think it is illegal to ban plastic items, effectively placing a ban on the plastic ban. The court battle demonstrates how cities and states are increasingly clashing over the legality of the plastic ban. In Figure 4 below, bans on plastic and bans on bans are labeled on the U.S. map [12].

Although it is possible to find other countries to export these plastic wastes temporarily or place plastic bans on these wastes, an efficient and effective process that eliminates plastic wastes is still needed. Incineration is the most mature method of reducing the amount of plastic waste, and it can accept mixed plastics. However, it generates carcinogenic products and releases environmental pollutions [13]. Mechanical recycling is a widely used recycling method, but it usually causes polymer degradation and cannot handle mixed plastics because of the immiscibility of polymer blends [14]. Additionally, it is an energy-intensive process [15]. Therefore, the conversion of plastics into useful products (e.g., fuels) will be a long-term solution to the plastic pollution problem.

### 1.2. Global Actions for Zero Plastic Pollution

To avoid a massive accumulation of plastic in the environment, coordinated global actions are urgently needed to reduce plastic consumption, increase rates of reuse, collection and recycling, expand safe disposal systems, and accelerate innovation in the plastic value chain. The focus of plastic pollution reduction strategies can be broadly partitioned into upstream (pre-consumption, such as reducing demand) and downstream (post-consumption, such as collection and recycling) measures [16]. Lau et al. [16] have applied a modeling approach to estimate mitigation potential under various intervention scenarios so an effective global strategy can be designed to appreciably reduce plastic pollution. According to their results, compared with “Business as usual”, the baseline scenario, the annual combined terrestrial and aquatic plastic pollution rates were reduced by 59% [17] under the Reduce and Substitute scenario, by 57% [18] under the Collect and Dispose scenario, and by 45% [19] under the Recycling scenario in 2040. However, neither pre- nor post-consumption interventions alone are sufficient to address the plastic problem. In the System Change scenario, which combines the maximum foreseen application of pre- and post-consumption solutions, the annual combined terrestrial and aquatic plastic pollution decreased by 78% [20] relative to BAU in 2040, which underlines that urgent extensive interventions are needed. 

Despite a considerable reduction in annual plastic production that can be effectively managed under the best-case System Change scenario, a substantial amount of plastic waste remained mismanaged. Mismanaged plastic wastes are the materials at high risk of entering the ocean via wind or tidal transport, or being carried to coastlines from inland waterways due to the lack of proper collection, sorting, recycling, or safe disposal. Fortunately, besides the current behaviors and new legislation, more and more emerging innovations and techniques will pave the way for a significant reduction of plastic wastes. 

## 2. Polymer Depolymerization Methods 

There are various types of plastics, and their variations allow proper material selection for a particular application. The most commonly used plastics materials are those with Society of the Plastics Industry (SPI) (see Table 1), including polypropylene (PP), polyethylene (high density, HDPE, and low density, LDPE), polyvinyl chloride (PVC), polystyrene (PS), and polyethylene terephthalate (PET) [21]. About 50%–70% of the total plastic waste is packaging materials derived from polyethylene, polypropylene, polystyrene, and polyvinyl chloride [22]. On average, polyethylene makes up the greatest fraction of all plastic wastes (69%), especially plastic bags, and polyethylene comprises 63% of the total packaging waste [23]. Therefore, polyolefins, including PP, HDPE, and LDPE, are the main group of synthetic plastics with the need for breakdown. 

Aside from the challenge of plastic waste disposal, another global issue is the energy crisis. Transportation consumes one-third of the world’s energy. The main energy sources for transportation are non-renewable fossil fuels. Today, these fuels are being consumed at an unsustainably high rate all over the world, and the global supply of fossil fuels will be depleted within 50–100 years at the current rate of consumption [24]. Production of fuel from plastics can simultaneously solve the challenges of plastic waste management and increasing energy demand. The conversion of these plastic wastes to usable oil is a growing and important field of study that can potentially mitigate the energy crisis. For those plastics with oxygen, such as polyethylene terephthalate (PET), degradation to its monomer and then repolymerization to yield high-quality plastic is a suitable recycling approach. 

### 2.1. Depolymerization for SPI Code 1: Polyethylene Terephthalate (PET)

Polyethylene terephthalate (PET) is a widely used commodity-grade thermoplastic with high chemical and impact resistance at room temperature. It is most commonly recognized for its use in injection-molded consumer packaging products such as water and soft drink bottles. Although PET does not pose a specific chemical threat to the environment, the growth of plastic discards such as water bottles into landfills causes a major environmental pollution problem. Besides, the recycling of PET aids in the conservation of raw petrochemical products and energy [26]. 

Two major conventional methods of recycling postconsumer PET exist: mechanical recycling and chemical recycling. Although it is economically viable relative to chemical recycling, mechanical recycling often produces new materials with lower quality which are not suitable for reuse in most beverage and food packaging owing to polymer degradation during processing and the high decontamination requirements. As a consequence, mechanically-recycled PET usually ends up in products such as fibers and engineering resins [27]. Chemical recycling of PET can guarantee the quality of the repolymerized products. Industrial processes for chemical recycling usually involve cleavage of the functional ester groups by reagents such as glycols (glycolysis), methanol (methanolysis), and water (hydrolysis), which are generally conducted at high temperature in the presence of catalysts such as manganese acetate [28], cobalt acetate [29], acetic acid, lithium hydroxide, sodium/potassium sulfate [30], and titanium (IV) n-butoxide [31]. Due to the unfavorable economics relative to mechanical recycling and low cost of starting monomer, chemical recycling of PET is not widely practiced. Thus, developing an environmentally safe, economically feasible, and industrially applicable chemical recycling process of PET is the goal for wide-scale applications. 

Many alternative catalysts have been studied to improve the outcomes of PET depolymerization (see Figure 5): Kamber et al. [26] studied a potent organic catalyst, N-heterocyclic carbenes, generated in situ from an ionic liquid in the presence of a base and is used for the transesterification reaction of PET with ethylene glycol in refluxing anhydrous tetrahydrofuran to generate bis(2-hydroxyethyl) terephthalate (BHET). The relatively mild reaction conditions and shortened reaction time of 1 h are attractive. Fukushima et al. [32] developed an organic phase transfer catalyst, guanidine 1,5,7-Triazabicyclo[4.4.0]dec5-ene (TBD), which is a potent neutral base and an efficient catalyst for the glycolysis of PET to its monomer BHET. Postconsumer PET beverage bottles were converted into BHET in 78% isolated yield with 0.7 wt% of TBD and excess amount of ethylene glycol at 190ºC for 3.5 h under atmospheric pressure. The recycling of the catalyst for more than five cycles was also demonstrated. Nunes et al. [33] demonstrated PET was successfully depolymerized to diethylterephthalate (DET) under supercritical ethanol with 1-Butyl-3-methylimidazolium tetrafluoroborate ([Bmim][BF_4_]) as catalyst. A robust PET conversion of 98 wt% was obtained, and adding [Bmim][BF_4_] can reduce the reaction time from ca. 6 h to 45 min.

### 2.2. Depolymerization for SPI Code 2, 4, and 5: Polyolefins

Polyolefins, including high-density polyethylene (HDPE), low-density polyethylene (LDPE), and polypropylene (PP), are the main group of synthetic plastics and by far the most important commodity polymers. In 2019, the production of polyolefin fiber reaches up to 1135 thousand metric tons in the United States [34]. Due to the fact that all atoms of PE and PP are connected through strong single C-C and C-H bonds, the chemical inertness of polyolefins sets a challenge for their depolymerization by low energy processes [35]. Due to the absence of oxygen and higher carbon and hydrogen content, the plastic fuels generated from polyolefins have the following advantages: avoidance of the need to upgrade further, non-acidity and non-corrosivity. The absence of water in plastic fuels leads to a very high higher heating value (HHV) [36,37]. Therefore, the fuels produced from polyolefins have fuel properties similar to fossil fuels and eventually can become an alternative energy source [38].

Two major methods have been applied to convert polyolefins into fuels: pyrolysis and supercritical water depolymerization. Pyrolysis can reduce most of the polymers including polyolefins. The pressure in the pyrolysis process is lower than the critical point, so water vapor is used for the conversion instead of supercritical water. However, there are some drawbacks to the pyrolysis of plastic wastes. It typically requires more energy than supercritical water depolymerization since a phase change is involved [39], and the process is operated at high temperature (450–800 °C) [40]. Also, this process has low oil yields without catalysts, and the oil generated from pyrolysis needs upgrading for fuel applications, resulting in a high processing cost [41]. In order to improve conversion, improve fuel quality, increase selectivity, and lower the pyrolysis temperature and reaction time, catalysts are added to reactions [42]. The acidic nature of most of the catalysts used enhances conversion by protonating the defective sites of polymers forming on-chain carbonium ions [43]. Selectivity and fuel quality vary with the strength of the catalyst’s acidity. Acid catalysts with meso- and micropores give a higher conversion. Primary cracking takes place in the macroporous surface, and further cracking is enhanced by micropores once the polymer is cracked [36]. The use of a strong catalyst results in the production of lower hydrocarbons ranging between C_3_ and C_5_. The catalysts used for polyolefins depolymerization can be categorized into several groups: fluid cracking catalysts, reforming catalysts, and activated carbon [43]. Fluid cracking catalysts include zeolite, silica-alumina, and clay. Reforming catalysts include transition metals loaded in silica-alumina. Activated carbon is also widely used and can be loaded with or without transition metals. The results of various investigations of polyolefins pyrolysis are listed in Table 2. The main effects of catalyst addition in pyrolysis are as follows: (1) the pyrolysis temperature for achieving a certain conversion is reduced drastically; (2) more iso-alkanes and aromatics in the C_5_–C_10_ range can be produced which are highly desirable gasoline-range hydrocarbons; and (3) the reaction rate is increased significantly [44].

Another method of polyolefin depolymerization, supercritical water depolymerization, is a thermochemical process, which usually requires moderate temperatures (≥374.15 °C) and pressures (≥22.129 MPa). As the reaction conditions approach the critical point of water, its properties such as dielectric constant, ionic strength, density, and heat/mass transport coefficients are changed rapidly. Especially, the rapid change of density correlates with other macroscopic properties to reflect changes at the molecular level, such as solvation power, molecular diffusivity, and viscosity [39]. These significant changes enable the supercritical water to give rise to fast, selective, and efficient reactions to convert organic wastes to oil, as compared to conventional depolymerization methods [52,53]. Watanabe et al. [54] used supercritical water for the conversion of low-density polyethylene (LDPE). The conversion of polyethylene is around 30% in the following conditions: temperature of 400 °C, pressure of above 30 MPa, and reaction time of 30 min. The effect of supercritical water on the depolymerization of polyethylene was explained based on the mechanism of H abstraction and β-scission reactions, which are shown below [54]:(1)$$Habstraction:\;R_{i}+M\overset{k_{\beta }}{\to }R_{i}H+R_{1}$$
(2)$$\beta -scission:\;R_{1}\overset{k_{\beta }}{\to }R_{i}+Ol$$
where *R_i_* is alkyl radicals; *M* is original n-alkane to decompose; *R_i_H* is n-alkane; *R*_1_ is the alkyl radical of mother n-alkane; *Ol* is 1-alkane; *k_H_* and *k_β_* are respectively rate constants for H abstraction and *β*-scission. In supercritical water depolymerization, as the molten polyethylene is diluted with dissolved water, the contribution of β-scission increases. This causes a shift of product distribution toward shorter chain hydrocarbon and an increase in 1-alkene yield. Moriya et al. [55] used supercritical water to convert high-density polyethylene (HDPE). They claimed oil yields of 90.2 wt% and 77.7 wt% after reaction times of 120 and 180 min in the conditions of 425 °C temperature and 42 MPa pressure. The yield in supercritical water is high, and coke production is small compared to the thermal cracking process. However, they stated the decomposition of PE proceeds slower in supercritical water depolymerization than in conventional thermal depolymerization.

Recently, Chen et al. [56] converted polypropylene (PP) into oil using supercritical water. The experiments were operated in the following conditions: 380–500 °C and 23 MPa over a reaction time of 0.5–6 h, and up to 91 wt% of PP was converted into oil at 425 °C with a 2–4 h reaction time or at 450 °C with a 0.5–1 h reaction time. They also claimed higher reaction temperatures (>450 °C) or longer reaction times (>4 h) lead to more gas products. About 80–90 wt% of the oil components have the same boiling point range as naphtha (C_5_–C_11_) and heating values of 48–49 MJ/kg. This conversion process is net-energy positive and has higher energy efficiency and lower greenhouse gas emissions than incineration, mechanical recycling, and pyrolysis. Therefore, the oil derived from PP has the potential to be used as gasoline blendstocks or feedstocks for other chemicals. Additionally, these researchers summarized the potential reaction pathways of major intermediates in the conversion process, as shown in Figure 6. At the temperature of 425 °C, PP was quickly decomposed into oligomers at short times (<0.5 h). When the reaction time further increases (from 0.5 to 4 h), a majority of the unsaturated aliphatics would be transformed into cyclics via cyclization. During the same period, small amounts of unsaturated aliphatics (olefins) may become saturated aliphatics (paraffins) and aromatics. Theoretically, aromatization can occur by either dehydrogenation of cyclics or cyclotrimetization of unsaturated aliphatics (olefins) [57].

### 2.3. Depolymerization for SPI Code 3: Polyvinyl Chloride (PVC)

Polyvinyl chloride (PVC) has the unique chemical property of being highly stable both in its chemical composition and its behavior when heated, and it has the potential to exhibit a wide variety of plastic–elastic properties from flexible to rigid PVC products when it is mixed with plasticizer and additives [58]. PVC contributes to ~12% of the total demand of plastics; 44.3 million metric tons of PVC was globally produced in 2018 [59]. However, the additives used in PVC make it one of the most problematic plastics for the environment because it releases phthalate plasticizers and chlorine-containing organics upon degradation.

The main problem with PVC recycling is the release of HCl, which leads to equipment corrosion. Even small quantities of PVC can contaminate entire batches of polymers in recycling plants and corrode reactors. This problem is circumvented by subjecting mixtures to a pretreatment (typically performed at 300 °C for 60 min) that reduces the chlorine content by ~75 wt% [60]. Dechlorination is then followed by catalytic pyrolysis of these materials. Another solution to this problem is the employment of HCl scavengers. A planetary ball mill grinds PVC with the HCl scavenger CaO in a process that does not require heat and affords a calcium salt by-product that can be washed away [58]. There remains high needs for better methods, catalysts, and HCl inhibitors to recycle PVC waste effectively. Inhibitors should hinder HCl formation, and catalysts should have the efficacy to facilitate PVC breakdown primarily to monomers in the presence of chlorine, HCl, or other plastics.

### 2.4. Depolymerization for SPI Code 6: Polystyrene (PS)

Polystyrene is an important class of materials widely used for the manufacture of packaging materials. It can be solid or foamed. General-purpose polystyrene is clear, hard and brittle, and it is in a solid state at room temperature. However, it flows if heated above the glass transition temperature (100 °C), and becomes rigid again when cooled. This temperature behavior is exploited for extruded polystyrene foam (Styrofoam), a light, waterproof material. Expanded polystyrene (EPS) is another similar foam material, which has been used as insulation, life vests and rafts, and food containers. However, the high degradation stability and low density of polystyrene cause major problems when it is disposed of in landfills [61]. Sorting polystyrene is also a challenge task since PS is usually mixed with other types of plastic or materials in the packaging. Thus, the processing of waste polystyrene is a significant issue.

The conventional methods of processing of polymeric wastes, like landfills and incineration, have many major drawbacks. The most common approach to recycling polystyrene is thermal or thermocatalytic decomposition to generate liquid oil, consisting mainly of C_6_~C_12_ aromatic hydrocarbons, a gaseous fraction, and a solid residue. The major disadvantages of the chemical methods for the processing of PS is the formation of a liquid product containing a wide range of different hydrocarbons. Due to the high styrene and α-methylstyrene contents of the produced liquid oil, its thermal-oxidative stability is very low. In addition, a high content of aromatic hydrocarbons in the liquid product can cause carbon formation problems in the engine when it is used as an automotive fuel. Although the use of catalysts in PS decomposition is helpful with the reduction of olefinic hydrocarbons, it significantly increases the yield of gases and leads to very intense coke formation during the depolymerization process [62]. Several factors inhibit the formation of styrene: heat transfer problems due to the difficulty of establishing contact between PS and the heat transfer material causing uneven heat supply, and intensified side reactions at high temperatures and long contact times.

One approach to overcome the issues mentioned above is depolymerization with a temperature of less than 550 °C and a vapor resident time of less than 10 s in a uniform heat distributed fluidized bed reactor. Liu et al. have reported styrene yields of 72~79% using this method [63]. Another approach is depolymerization in a hydrocarbon medium, which can avoid the heat transfer problem and side reactions mentioned above and achieve high styrene selectivity. Several media for the depolymerization of PS have been studied. Table 3 lists the yields of styrene using various hydrocarbon media.

A high yield of monomer during the depolymerization reaction in a hydrocarbon medium suggests that a number of side reactions that would lead to a decrease in styrene selectivity were suppressed under the conditions used. The generalized scheme of polystyrene transformations based on the obtained experimental data is shown in Figure 7. The studies demonstrated that the high selectivity toward styrene can be explained by a change in the depolymerization mechanism during the reaction in hydrocarbon media. It was first established that the direct formation of styrene dimers from a fragment of a polymer molecule in the hydrocarbon medium does not proceed; the dimers can be formed only via secondary reactions [61].

### 2.5. Depolymerization for SPI Code 7: Other Polymers

The #7 category was designed as a catch-all for polycarbonate (PC) and “other” plastics, such as polyesters, polyethers, nylon (or polyamide), polymethyl methacrylate (PMMA), acrylonitrile butadiene styrene (ABS), etc., so reuse and recycling protocols are not standardized within this category. Several studies have been performed to search for efficient methods to depolymerize them.

#### 2.5.1. Polycarbonates (PC), Polyesters and Polyethers

Polycarbonates, polyesters, and polyethers are three major groups in the SPI code #7 category. They respectively contain carbonate group, ester functional group, and ether linkages in their main chain, and similar chemical recycling methods can be applied on them. Cantat [66] and his group have depolymerized polycarbonates, polyesters, and polyethers using hydrosilanes as reductants and metal-free catalysts to generate functional chemicals such as alcohols and phenols at room temperature. They also reported the depolymerization of polyesters (such as polylactic acid, or PLA) in the presence of hydrosilanes with the cationic pincer complex as catalyst toward the formation of silyl ethers or the corresponding alkanes under mild conditions [67]. Liu et al. [68] reported a series of imidazole-anion-derived ionic liquid were facilely synthesized and used for efficient catalyzing alcoholysis of polyester wastes, such as PLA and polyhydroxybutyrate (PHB). Robertson et al. [69] used Ruthenium(II) PNN complexes to depolymerize many polyesters into diols and polycarbonates into glycols plus methanol via hydrogenation. Grewell [70] and his team presented their study of depolymerization of PLA to lactic acid or lactic esters with yields of over 90% by heating at a temperature of 50~60 °C in the medium of water, ethanol, or methanol with the catalysts of various carbonate salts and alkaline metal oxides.

#### 2.5.2. Nylon (or Polyamide)

Nylon is a family of synthetic thermoplastic, and nylon-6 is formed by ring-opening polymerization unlike most of the nylons, which gives it unique properties. Kamimura [71] and his team have depolymerized nylon-6 in several ionic liquids (emim BF4, PP13 TFSI, TMPA TFSI, etc.) at 300 °C to give corresponding monomers in good yields of 35~86%. Polyphthalamide (PPA) is a subset of thermoplastic synthetic resins in the polyamide (nylone) family. Phillips [72] and his team have reported a strategy for designing stimuli-responsive patterned PPA plastics that are capable of responding to chemical signals in the environment by changing shape and depolymerizing once the signal reacts with the trigger. Moore et al. [73] illustrated PPA undergoes mechanically initiated depolymerization to revert the material to monomers using a heterolytic scission mechanism, and the obtained monomer was repolymerized by a chemical initiator, effectively completing a depolymerization–repolymerization cycle.

#### 2.5.3. Polymethyl Methacrylate (PMMA)

PMMA, also known as acrylic, is a transparent thermoplastic, although it is often technically classified as a type of glass. Godiya et al. [74] studied thermal pyrolysis of PMMA to produce its monomer methyl methacrylate in high yield. Ouchi [75] and his group studied the depolymerization of a chlorine-capped polymethyl methacrylate (PMMA-Cl) at temperatures of >100 °C.

#### 2.5.4. Acrylonitrile Butadiene Styrene (ABS)

ABS is a common thermoplastic. It is amorphous, and therefore has no true melting point. Advantageous functional and processing properties of the ABS co-polymer and its combinations make it a material ideal for the manufacturing of toys and electronic equipment. However, after a short period of use, most of them lose their functional properties and constitute a waste product. Marciniak [76] and his team present the application of the material obtained from the recycling of ABS to produce filament using Fused Filament Fabrication (FFF) after regranulation, which allows the possibility of recycling of the operating parts of 3D printers.

## 3. Challenge and Future

Despite their problems, plastics provide benefits to our daily lives, and there are no alternatives for immediate deployment at global scales. Therefore, to stop the waste flow into the ecosystem, a roadmap for the development of plastic industry is outlined: elimination, innovation, and circularity, which was suggested by EU Action Plan, the Global Plastic Action Partnership, and the Ellen MacArthur Foundation [77].

### 3.1. Elimination

Elimination of all problematic and unnecessary plastic items is the first goal on the roadmap, and it will entail a dramatic reduction in the number of plastic formulations from thousands of types of plastics found on the market. However, not all formulations are easily recycled, which adds a level of complexity. The seven categories mentioned above are only a small portion of the story. In most plastic products, the basic polymer is usually combined with different additives to improve the performance, functionality, and aging properties of the basic polymer [4]. Additives complicate recycling, since they must be identified, separated, and properly disposed of in order to recycle a post-use plastic back into virgin resin. Therefore, reducing the number of additives used will require compromise by consumers and producers. Finding leadership on this issue will be a vital first step [78].

### 3.2. Innovation

Innovation that ensures the plastics needed are useable, recyclable, or compostable is the next goal on the roadmap. The science is still developing in this field, but chemical recycling is a field of study that should be explored. For polyolefin waste management, supercritical water depolymerization can be a good option since it possesses the following advantages: (1) moderate operation temperature and pressure; (2) no catalyst needed during the process; and (3) high-quality and valuable products [56,79,80]. For the other types of polymers, hydrothermal catalytic depolymerization or depolymerization with ionic liquids is still the most suitable choice. In the future, development of techniques and catalysts to efficiently recycle these polymers will be a significant task.

### 3.3. Circularity

Circularity of all the plastic items used to keep them in the economy and out of the environment is the final goal. Therefore, design for circularity is an emerging field that will require manufacturers to rethink how products are designed. Reusing, recycling, and remanufacturing principles must be applied during the product design phase [81].

Above all, a combination of international policy and consumer demand for change will necessarily slow the accumulation of mismanaged plastic wastes. Both challenges and opportunities will make the next 100 years of plastics manufacture and use significantly different.

## Figures and Tables

**Figure 1 polymers-13-00449-f001:**
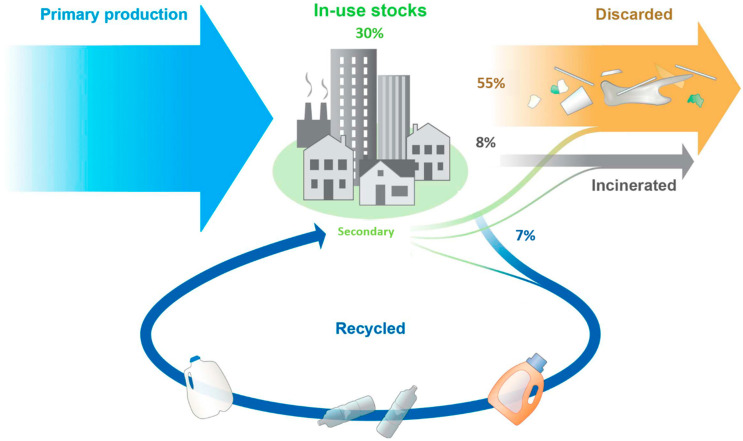
Global production, use, and fate of polymer resins, synthetic fibers, and additives (1950 to 2015) (adapted from [3]).

**Figure 2 polymers-13-00449-f002:**
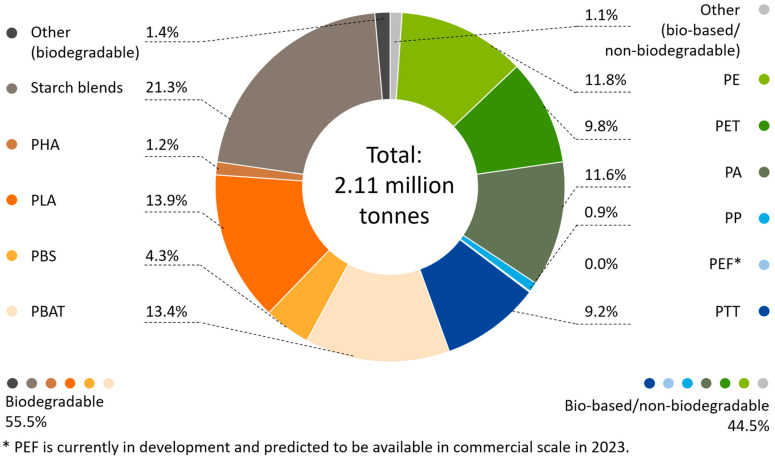
Global production capacities of bioplastics 2019 (by material type) (adapted from [7]).

**Figure 3 polymers-13-00449-f003:**
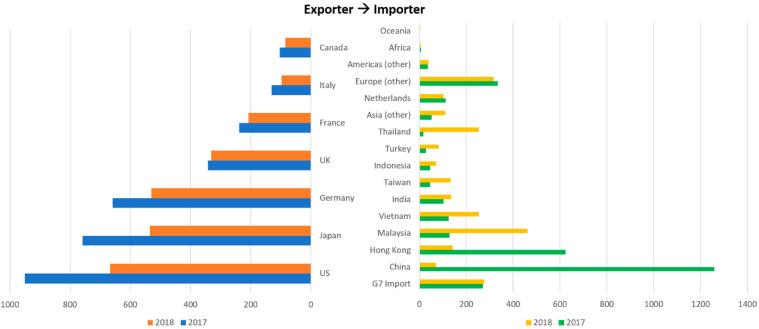
Comparison of exports of plastic waste, pairings and scrap from G7 countries in the first half of 2017 and 2018 (in kilo tons) (adapted from [11]).

**Figure 4 polymers-13-00449-f004:**
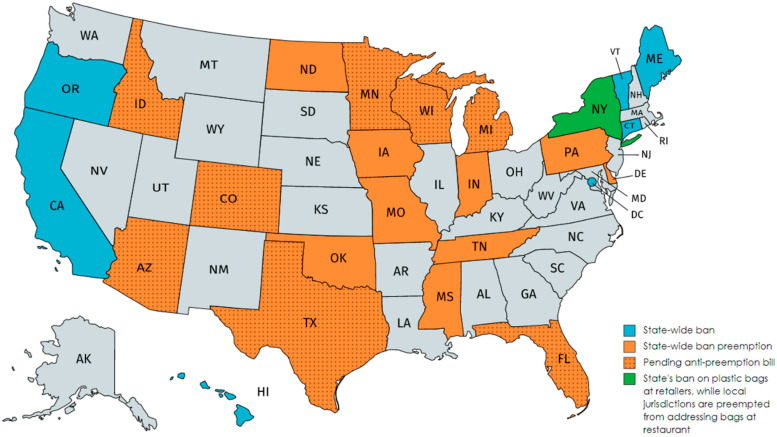
U.S. map with the labels of bans on plastic and bans on bans (adapted from [12]).

**Figure 5 polymers-13-00449-f005:**
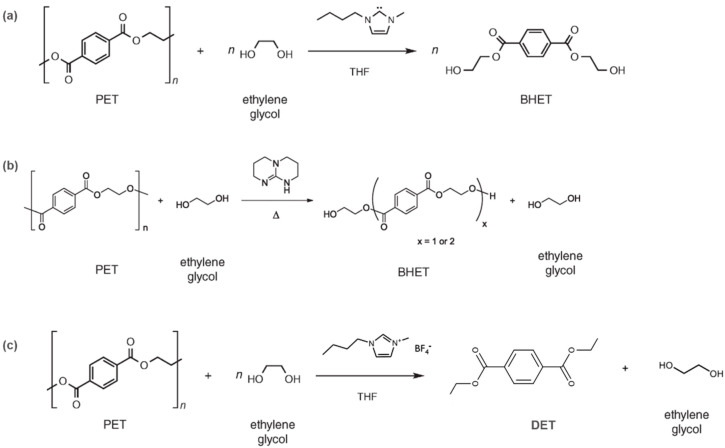
Reaction scheme of the depolymerization of polyethylene terephthalate (PET) using (**a**) N-heterocyclic carbenes [26], (**b**) guanidine 1,5,7-Triazabicyclo[4.4.0]dec-5-ene (TBD) [32], and (**c**) 1-Butyl-3-methylimidazolium tetrafluoroborate ([Bmim][BF_4_]) [33].

**Figure 6 polymers-13-00449-f006:**
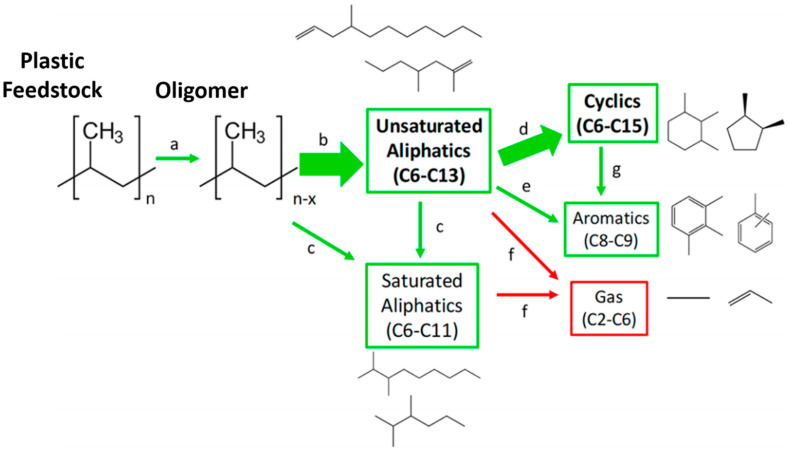
Potential reaction scheme of converting polypropylene via supercritical water depolymerization process. A green box represents oil phase products and a red box denotes gas products. The thickness of arrows represents the relative amounts of products [56].

**Figure 7 polymers-13-00449-f007:**
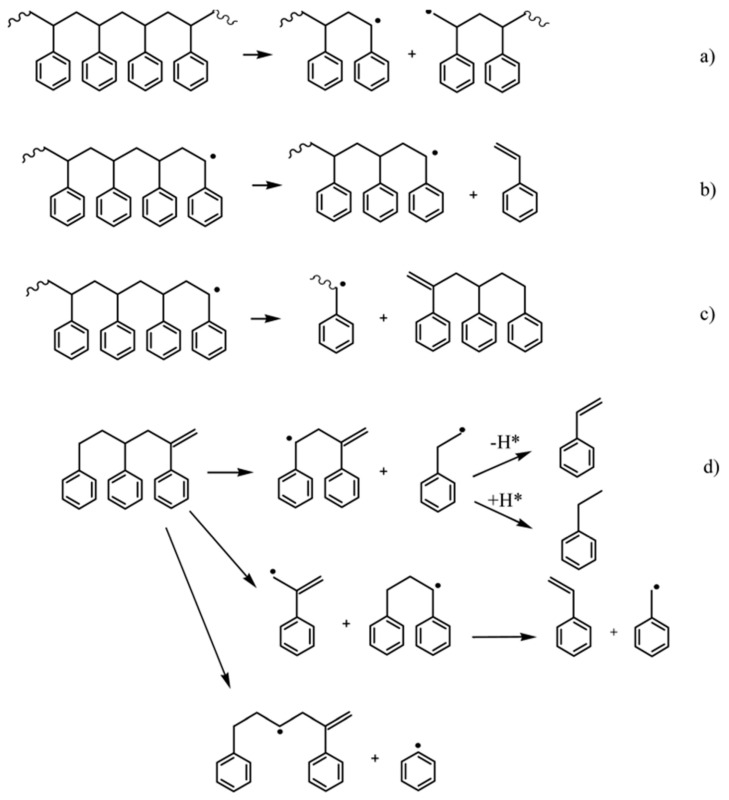
Scheme of the reactions occurring during polystyrene depolymerization in the hydrocarbon media: (**a**) hemolytic cleavage of the C-C bond, (**b**) β-cleavage of the C-C bond accompanied by the formation of styrene, (**c**) 1,7-hydrogen transfer accompanied by the formation of trimer, and (**d**) cracking of the trimer [61].

**Table 1 polymers-13-00449-t001:** List of plastics with Society of the Plastics Industry (SPI), their properties, uses and recycling situations.

SPI Number	Full Name	Chemical Structure	Uses	Currently Recyclable? [25]
1	Polyethylene terephthalate	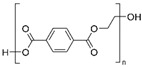	Disposable bottles, medicines, etc.	Yes
2	High-density polyethylene	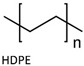	Durable containers	Yes
3	Polyvinyl chloride	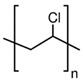	Piping, cables, garden furniture, fencing, carpet backing	No
4	Low-density polyethylene	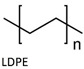	Plastic bags, wrapping films, trays, computer components	Mostly no
5	Polypropylene	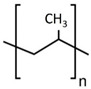	Bottle caps, reusable food containers, car parts	Sometimes
6	Polystyrene	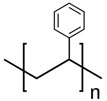	Plastic utensils, packaging peanuts, styrofoam	Sometimes
7	Other: for example, polycarbonate, polymethyl methacrylate	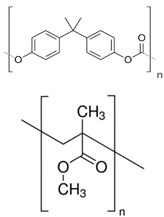	Multilayer barrier films, toothbrushes, CDs and DVDs	No

**Table 2 polymers-13-00449-t002:** Mass balance of crude oil, residue and gas yields on pyrolysis of polyolefins using various temperatures and catalysts.

Plastics	Temp. (°C)	Catalysts	Oil (wt%)	Gas (wt%)	Tar (wt%)	Residue (wt%)	Investigation
HDPE	260	-	3.0	0.0	15.0	82.0	Ishihara et al. [45]
HDPE	260	NaY zeolite	38.0	10.0	36.0	16.0	Ishihara et al. [45]
HDPE	400	Silica-alumina	93.0	7.0		0.0	Beltrame et al. [46]
HDPE	400	H-Y zeolite	91.0	9.0		0.0	Beltrame et al. [46]
HDPE	400	-	44.0	20.0	13.0	17.0	Ohkita et al. [47]
HDPE	400	HZSM-5 zeolite	45.0	50.0	1.0	Trace	Ohkita et al. [47]
HDPE	420–440	-	74.0	9.0		17.0	Sharma et al. [38]
HDPE	500	-	93.0	7.0		0.0	Williams et al. [48]
HDPE	600	-	29.0	6.0		65.0	Beltrame et al. [45]
HDPE	760	-	42.4	55.8		1.8	Buekens et al. [43]
LDPE	350	-	15.0	34.0		46.0	Morid et al. [49]
LDPE	350	H-mordenite	32.0	43.0		25.0	Morid et al. [49]
PP	220	Silica-alumina	46.0	12.0	36.0	5.0	Ishihara et al. [50]
PP	260	Silica-alumina	32.0	6.0	29.0	33.0	Ishihara et al. [50]
PP	350	H-mordenite	24.0	43.0		15.0	Morid et al. [51]
PP	380	-	64.9	24.7		10.4	Ohkita et al. [46]
PP	380	Silica-alumina	68.8	24.8		6.4	Ohkita et al. [46]
PP	500	-	95.0	5.0		0.0	Williams et al. [47]
PP	740	-	48.8	49.6		1.6	Buekens et al. [43]

**Table 3 polymers-13-00449-t003:** Yields of Styrene using various hydrocarbon media.

Hydrocarbon Media	Temperature (°C)	Vapor Residence Time (s)	Styrene (wt%)	Investigation
benzene	550	3~9	75.6	de la Puente et al. [64]
fluid catalytic cracking light cycle oil	550	3	55.4	Arandes et al. [65]
light cycle oil	500~550	1.6~2.6	84.4	Dement’ev et al. [61]
heavy cycle oil	500~550	1.6~2.6	82.5	Dement’ev et al. [61]

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
