# Peer review of "Current Technologies in Depolymerization Process and the Road Ahead"

_polymers, 2021, doi:10.3390/polym13030449_

Round 1

Reviewer 1 Report

REVIEW:
The paper is quite interesting, since it presents a worldwide problem related with the plastics pollution and possible solutions to solve the problem focused on different plastic depolymerization methods in order to convert waste plastics into fuels to eliminate the harm to the environment and decrease the dependence on fossil fuels. As mentioned in the last point of the paper (point 3. Challenge and Future), supercritical water depolymerization can be a good option for polyolefin waste management, however it is necessary to develop techniques to efficiently recycle other kinds of polymers. 

ENGLISH WRITING:
The paper's readability is very good.

TYPOS:
Line 315: Table 2 must be renumbered to Table 3.

Author Response

Reviewer #1

REVIEW:
The paper is quite interesting, since it presents a worldwide problem related with the plastics pollution and possible solutions to solve the problem focused on different plastic depolymerization methods in order to convert waste plastics into fuels to eliminate the harm to the environment and decrease the dependence on fossil fuels. As mentioned in the last point of the paper (point 3. Challenge and Future), supercritical water depolymerization can be a good option for polyolefin waste management, however it is necessary to develop techniques to efficiently recycle other kinds of polymers. 

More studies on the depolymerization of polymers in the #7 category have been added, and all these studies have been divided into four groups: polycarbonates, polyesters and polyethers, nylon, PMMA, and ABS. Please see the updated manuscript.

TYPOS:
Line 315: Table 2 must be renumbered to Table 3.

Revised as requested.

Reviewer 2 Report

Reviewers' comments:

Manuscript ID: polymers-1089022

Title: Current Technologies in Depolymerization Process and The Road Ahead.

Manuscript Type: Review.

Reviewers' comments:

The manuscript describes the Current Technologies in Depolymerization Process and The Road Ahead. The manuscript needs a detailed editing. Some markings are made to just illustrate the extent of editing needed. A thorough revision addressing all the concerns is needed and if the authors are prepared to do that it can be considered for a review of the revised manuscript.

The authors need to consider the following comments

- Abstract must be improved.

- Keywords must be improved.

- The introduction section should be improved; more related papers must be discussed and superiority, novelty, critical improvement in this study must be clarified.

- I think more references should be included. 64 is not enough for a review paper. Besides, please have more recent publications.

- 2.5. Depolymerization for SPI Code 7: Other Polymers - must be improved.

- Conclusions: the authors need to improve with more specific results and conclusions, i.e. academic novelty or technical advantages.

- Author should provide the references for all equations and formula.

- References: Make all references in same format for volume number, page number and journal name, because it is difficult to searching and reading.

- Furthermore, they should add the graphical abstract, it is use full to readers.

Author Response

Thanks for the comments and suggestions. 

- Abstract must be improved.

The abstract is modified. Please see the changes as highlighted below:

Although plastic is considered an indispensable commodity, plastic pollution is a major concern around the world due to its rapid accumulation rate, complexity, and lack of management. Some political policies, such as the Chinese import ban on plastic waste, force us to think about a long-term solution to eliminate plastic wastes. Converting waste plastics into liquid and gaseousfuels is considered a promising technique to eliminate the harm to the environment and decrease the dependence on fossil fuels, and waste plastic recycling by converting them into monomers is another effective solution to plastic pollution problem. This paper presents the critical situation of plastic pollution, various methods of plastic depolymerization based on different kinds of polymers defined in the Society of the Plastics Industry (SPI) Resin Identification Coding System, and the opportunities and challenges in the future.

- Keywords must be improved.

Keywords are revised as requested. The new ones are Plastic pollution; Plastic depolymerization and recycling; Plastic depolymerization challenges.

- The introduction section should be improved; more related papers must be discussed and superiority, novelty, critical improvement in this study must be clarified.

A section named “1.2 Global Actions for Zero Plastic Pollution” has been added at the end of Introduction section using the modeling approach to state how to deal with plastic pollution and introduce the next section of depolymerization methods.

- I think more references should be included. 64 is not enough for a review paper. Besides, please have more recent publications.

Another 19 references are added in the manuscript. Almost all of new papers were published since 2010, and most of are within the last 3 years. 

- 2.5. Depolymerization for SPI Code 7: Other Polymers - must be improved.

More studies on the depolymerization of polymers in the #7 category have been added, and all these studies have been divided into four groups: polycarbonates, polyesters and polyethers, nylon, PMMA, and ABS. Please see the updated manuscript.

- Conclusions: the authors need to improve with more specific results and conclusions, i.e. academic novelty or technical advantages.

The authors have rewritten this section by adding a novel perspective of development roadmap for plastic industry suggested by MacArthur Foundation. Please see the updated manuscript.

- Author should provide the references for all equations and formula.

Revised as requested.

- References: Make all references in same format for volume number, page number and journal name, because it is difficult to searching and reading.

Revised as requested.

- Furthermore, they should add the graphical abstract, it is use full to readers.

A graphical abstract has sent to the editor and is attached with this reply. 

This manuscript is a resubmission of an earlier submission. The following is a list of the peer review reports and author responses from that submission.